# Preparation of Novel Graphene/Silicone Rubber Nanocomposite Dielectric Foams

**DOI:** 10.3390/polym14163273

**Published:** 2022-08-11

**Authors:** Fei Jia, Cong Liu, Bo Yang, Alamusi Lee, Liangke Wu, Huiming Ning

**Affiliations:** 1Department of Astronautical Science and Mechanics, Harbin Institute of Technology, Harbin 150001, China; 2School of Manufacturing Science and Engineering, Southwest University of Science and Technology, Mianyang 621010, China; 3National Engineering Research Center for Technological Innovation Method and Tool, School of Mechanical Engineering, Hebei University of Technology, Tianjin 300401, China; 4College of Aerospace Engineering, Chongqing University, Chongqing 400044, China

**Keywords:** silicone rubber, nanocomposites, dielectric constant, foaming expansion rate

## Abstract

In order to obtain high dielectric silicone rubber (SR)-based nanocomposites, graphene (Gr) was added by ultrasonication and mechanical mixing for the preparation of a microporous structure. It was discovered that the Gr content and the expansion rate had a great impact on the cellular structure. Based on the effects of the Gr content and the expansion rate on the dielectric property, hybrid materials were prepared and better properties appeared, as expected. For all samples, the dielectric constant increased with the Gr content until 3 wt% and then decreased. When the Gr content was 3 wt% and the expansion rate was 2, the dielectric constant reached 18.14 (1 kHz), which was 55% higher than that of the non-expansion sample (11.74) and several times that of the pure sample (3~6). Meanwhile, the dielectric loss was less than 0.01. This work proposed a method for producing high dielectric materials with important applications in the field of capacitors, sensors, and micro-resistors.

## 1. Introduction

With the development of the modern electric industry, high dielectric materials play an important role in related fields such as capacitors, sensors, and memory devices [1]. Among them, silicone rubber modified by nanofillers has attracted much attention due to its outstanding electrical insulation and flexibility. In the composites, the connected conductive nanofillers and the polymer segments may form a dielectric pathway [2]. Silicone rubber foam is a widely used porous polymer material due to its combined characteristics of silicone rubbers and foam materials [3,4].

In recent decades, researchers have done much work on the addition of nanofillers to improve the dielectric property of silicone rubber or silicone rubber foam. Liu et al. [1] prepared silicone rubber/polyolefin elastomer (SR/POE) blends containing MWCNTs by the melt-blending and hot-pressing method. The composites showed a sea-island structure and the dielectric permittivity and loss increased significantly with increasing MWCNTs. It was attributed to the space charge carriers that aggregated at the interface between MWCNTs and the polymer and formed the interfacial polarization. Madidi et al. [5] studied the dielectric behavior and the electrical conductivity of room-temperature vulcanized silicone rubber SR/TiO_2_ nano- and micro-composites prepared by a combination of mechanical mixing and ultrasonication. 

It was observed that the dispersion of TiO_2_ can be improved by a surface treatment of TiO_2_ with a surfactant. The results showed that the relative permittivity increased with the TiO_2_ concentrations. Besides, compared with nanocomposites, the permittivity tended to increase more strongly, which may have been due to the relatively higher interfacial polarization in the microcomposite systems. Other nanofillers such as Al, copper calcium titanate (CCTO), reduced graphene oxide (rGO), and CNTs were also used [6,7,8,9]. Generally speaking, the addition of conductive nanofillers may have a positive effect on the improvement of dielectricity due to the constructed conductive pathway in the matrix [4,9,10]. Besides, extensive research has been done to improve SR foam properties. Song et al. [3] prepared silicone rubber foam by a foaming process with super carbon dioxide (SC CO_2_). It was found that the procuring time played an important role in the foaming process, which affected the cross-linked network structure. When the procuring time was set as 5 min, the samples reached the lowest density, and the best pore microstructure. SR/MWCNTs/Fe_3_O_4_ nanocomposite foams were prepared using a supercritical carbon foaming process [11]. It was found that the microwave-absorbing ability (SE = 27.5 dB in the frequency range of 8.2–12.4 GHz) was greatly improved owing to the existence of the cellular structure and magnetic Fe_3_O_4_ nanoparticles, thus showing the potential in the application as electromagnetic interference shield (EMIS) materials. Chen et al. [12] prepared a series of silicone foams with outstanding performances by direct ink writing. It had extreme compressibility, cyclic endurance, and stretchability. With a structure designed in 3D printing, the foam may be used to absorb oil selectively. Xiang et al. [13] prepared a series of microcellular silicone rubber foams with supercritical carbon dioxide. The influences of added silica (reinforcing agent) on the cellular morphology, nucleation, and the rheological behavior were studied. The composite containing 70 phr silica showed the optimal cell morphology, minimum cell diameter (708 nm), and highest cell density (1.02 × 10^11^ cells/cm^3^). Meanwhile, the calculated surface tension was also dramatically increased from 158.95 nN/m to 1092.74 nN/m. A constitutive model established based on the statistical theory of rubber elasticity of silicone foams by Wei et al. [14] was applied to fit the compressive stress–strain data. The fitting curves were generally inconsistent with the experimental data of both mono- and bi-modal foams.

However, to date, very few reports about the influences of nanofillers on the dielectric properties of silicone rubber foams have been published. Liu et al. [15] synthesized silicone foams with and without liquid fillers (silicone oils of various types and glycerol, respectively) and examined their mechanical and dielectric properties. Compared to dry silicone foams, the composites containing oil possessed a low Young’s modulus (31 kPa) and a relative dielectric permittivity around 5, and was a good candidate as the dielectric layers in capacitive sensors.

In this work, we prepared silicone rubber/graphene (SB/Gr) composite samples. The dielectric properties of the composite samples with graphene fillers increased dramatically and the dielectric loss was extremely low. Based on effects of the nanofiller contents and the expansion rate on the composites properties, a hybrid SR foam material was also prepared, which showed better properties, as expected. It is of great interest in the related fields.

## 2. Experiments

### Materials and Experiments

Two-component additional curable room temperature vulcanized silicon rubber (HY-F661~HY-F665, MW ≈ 15000~2000, Hong Ye Jie Tech. Co., Ltd., Shenzhen, China) and graphene (r (5~10 μm), 2–15 layers, purity 80 ± 0.5%, Morsh Co., Ltd., Ningbo, China) were all obtained commercially and used as received. The last number of the silicone rubber is the foaming expansion rate *α*, i.e., the foaming expansion rate of HY-F661 is 1 (non-foamed) and foaming expansion rate for HY-F665 is 5. Figure 1 shows an example for the preparation process of the silicone rubber foam/graphene (SR/Gr) composites, whose foaming expansion rate is 2. Uniform distribution of graphene in matrix is important, and sample preparation was divided into the following three main steps. The dispersion of graphene in ethanol (Step 1) and then the mixing of silicone rubber/graphene (Step 2) aimed at avoiding agglomeration.

Step 1: (a) Add proper graphene into ethanol and mix them by ultrasonication for 30 min; (b) dry the solution mixture in a vacuum oven at 120 °C for 10 min.

Step 2: (a) Cool down the mixture from Step 1 to room temperature; (b) add proper group A (HY-F662B) into it and mix them with a planetary machine (Thinky Mixer AR-100, Thinky Co., Ltd., Tokyo, Japan) for 5 min; (c) add a little KH550 into the mixture and mix them with the planetary machine for 10 min; (d) add group B (HY-F662A, mass A = mass B) and then mix them with an electric mixer for 30 s. 

Step 3: (a) Put the mixture into a cylindrical mold and cure (60 °C, 1 h) it in a vacuum chamber; (b) cut the samples into 10 mm cylinders, with a height of 2 mm.

In this study, the samples with α = 1, 2, 3, 4, 5 are denoted as F1, F2, F3, F4, and F5, respectively. Besides, the hybrid sample, which is denoted as F2–5, was also prepared for comparison. The materials of all samples are illustrated in Table 1.

## 3. Morphology

Using Image-Pro Plus 6.0 to deal with the image obtained by SEM (NEC 7610F), the dimension and number of foam cells were counted. The cell density $D_{\mathrm{cell}}$ was calculated with the following equation [16,17].
(1)$$D_{\mathrm{cell}}=\frac{1}{1-f}{\left(\frac{nM^{2}}{A}\right)}^{\frac{3}{2}}$$
where *n* is the number of bubbles in the micrograph, *A* is the micrograph area, and *M* is the magnification factor of the micrograph. *f* is the void fraction of the composite, which is calculated by
(2)$$f=\left(1-\frac{\rho_{f}}{\rho }\right)\times 100\%$$
where ρ is the density of the non-foam composite. $\rho_{f}$ denotes the density of the foam composite. From definition, $\alpha =\rho /\rho_{f}$. Substituting it into Equation (2) yields
(3)$$f=\left(1-\frac{1}{\alpha }\right)\times 100\%$$

The dielectric property was measured by broadband dielectric spectroscopy (Novocontrol GMBH, Hundsangen, Germany). The frequency ranged from 1 Hz to 10 MHz. Before the test, Cu electrodes were pasted on both sides of the samples with conductive silver paste.

## 4. Results and Discussion

### 4.1. Morphology

Figure 2 shows the graphene dispersion in the composites containing various contents of Gr. From Figure 2, it is clear that when the Gr content was less than 3 wt%, the fillers may have been dispersed in the matrix uniformly and separately. With the increase in Gr content, contacts between different fillers were observed.

Figure 3 shows the cell structure of the composites with various expansion rates and Gr contents. As shown in Figure 3, both open cells and closed cells existed in all samples. When the foaming expansion rate was 2, with the addition of Gr, the number of cells increased and the cell size became uniform. According to the cell formation mechanism [18], it was attributed to the increased heterogeneous nucleation agents resulting from the addition of Gr. Furthermore, when the Gr content reached 4.0 wt%, larger cells appeared, which was caused by the combination of neighboring cells.

In the samples containing 3.0 wt% Gr, the numbers of the foam cells as well as the closed cells increased with the foaming expansion rate. In samples F2–5, it included some large cells and more small cells.

To analyze the foam cell morphology, the relative density, cell size distribution, and density of the cells were counted. For example, the results of composites containing 1 wt% are shown in Table 2 and Figure 4. From Table 2, with increasing foaming expansion rate *α*, the average cell size decreased, and the cell number as well as the cell density increased. In the composites with a low expansion rate, the large size cells (>150 μm) dominated, whereas the small size cells occupied a higher percentage in the composites of a higher expansion rate. The average cell size and the cell density of F2–5 stood between those of F4 and F5. 

### 4.2. Dielectric Properties

Figure 5 is the dielectric constant (1000 Hz) of the composites with different foaming expansion rates. It is clear that for pure silicone rubber, the dielectric constant decreased with the increase in the foaming expansion rate. The generation of cells was attributed to the H_2_ gas generated in the vulcanization process, and thus the foaming occurred. The dielectric constant of the foamed samples $\varepsilon_{r}$ can be obtained by [19]
(4)$$\varepsilon_{r}=\varepsilon_{H}f+\left(1-f\right)\varepsilon_{SR},\;$$
in which $\varepsilon_{H}=1.00026$ and $\varepsilon_{SR}$ are the dielectric constants of H_2_ and silicone rubber, respectively.

For simplicity, let $\varepsilon_{H}=1$ and $\varepsilon_{SR}$ be taken as experimental $\varepsilon_{r}$ for f=0 in Table 3, and Equation (4) can be further written as
(5)$$\varepsilon_{r}=6.25-5.25f$$

The theoretical $\varepsilon_{r}$ listed in Table 3 is obtained from Equation (5). In comparison, the theoretical values, except for F5, are in good agreement with the experimental values for the pure SR materials. Noting that the dielectric constants of silicone rubber foams without Gr decreased monotonously with foaming expansion rate.

Referred to composites, the dielectric constant is usually described by the threshold theory [20] and tunnel effect [21]. To be specific, in discussion on the permittivity of the foaming composite materials, two factors should be considered: (1) the effect of the graphene and (2) the foaming expansion rate. 

In Figure 5, the dielectric constant of the porous samples with different expansion rates varying with graphene loading is illustrated for 1 kHz. When the expansion rate was given, the dielectric constant first increased and then decreased with graphene content. The reason is that Gr and foam cells form many microcapacitors. However, an overdose of nanofillers may form a conductive pathway composed of Gr [21], which can lead to a decrease in the dielectric constant. As depicted in Figure 5, there exists a threshold [20] of about 3 wt%, beyond which the dielectric constant begins to decrease with graphene loading. 

When the graphene content is constant, the dielectric constant of the porous samples containing low Gr content (<3 wt%) is enhanced due to the formation of microcapacitors, compared to the non-foamed samples, which is inconsistent with Equation (5). For example, when the frequency was 1 kHz and the Gr content was 3 wt%, the dielectric constant (18.14) of the sample with *α* = 2 was 55% higher than that (11.74) of the non-foam sample. When the threshold was reached, the increase in the expansion rate squeezed the existing space of the graphene, which easily led to the formation of conductive pathways composed of Gr. Therefore, the dielectric constant of the porous samples with high Gr content (>3 wt%) may have been worse than non-foam sample F1. Among the porous samples, the dielectric constant decreased with the expansion rate. The increase in the expansion rate led to a decrease in the graphene per unit volume. The dielectric constant depends on the amount of graphene per unit volume before the threshold is reached. The less graphene per unit volume, the fewer microcapacitors. 

Figure 6 illustrates the dielectric constant of the composites (3 wt% Gr) varying with frequencies. When *α* was given, the dielectric constant decreased with the frequency. Similar trends have been reported for other conductive composites [22,23,24]. For samples with the same Gr content, the dielectric constant decreased with the foaming expansion rate, which showed the same trend as the pure silicone rubber samples described as Equation (5). The dielectric constant of the material is dependent on its microstructure. From Figure 4 and Table 2, F2 had relatively few and larger foam cells and F5 had relatively more and smaller foam cells. Combining with the results shown in Figure 5, it is reasonable to believe that larger and more cells formed by its microstructure contribute to more effective microcapacitors, and further lead to dielectric constant enhancement. For a given sample of F1 to F5, it seems not possible to obtain larger and more cells at the same time. Compounding samples with different expansion rates yielded the hybrid sample F2–5, which had more optimally sized (100–150 μm) foam cells. As shown in Figure 6, the dielectric constant of F2–5 was higher than that of other samples with different foaming expansion rates, which validates the enhancement of the dielectric constant by the large cell and large number of cells.

Figure 7 depicts the dielectric loss of the composite samples with respect to the frequency for different foaming expansion rates. For a given frequency, the dielectric loss decreased as the expansion rate increased, which is opposite to the change of the dielectric constant. As shown in Figure 4 and Table 2, while the foaming expansion rate increased, the small cells predominated, which made the Gr more uniformly distributed, and thus the dielectric loss decreased. Obviously, the dielectric loss of F2–5 was lower than that of all other samples, and its dielectric constant was the highest, which may be another piece of evidence of the synergistic effects of the large cell and large number of cells. 

## 5. Conclusions

In this work, SR/Gr composites of various foaming expansion rates (1–5) were prepared. The effects of Gr contents and foaming expansion rates on the dielectric property were discussed. The results are concluded as follows: The foaming cells may have improved the dielectric properties of SR. With the increase in foaming expansion rate, the cell size decreased and the cell number increased. The small cells improved the tunneling effects, and thus the dielectric constant decreased. The cell number increment was beneficial to the uniform Gr distribution, which enhanced the dielectric constant. That is the reason why the dielectric constant was higher at foaming expansion rate (F2) while the performance was conversely at higher foaming expansion rates. The addition of low contents of Gr (≤3 wt%) may have improved the dielectric property, whereas more Gr may have decreased it. This is attributed to the enhancement of the polarization effect (≤3 wt%) and the formation of a conductive pathway (>3 wt%). In order to prepare composites with high dielectric factors (large cells, large cell number), F2–5 composites were fabricated. In the F2–5 samples, the large cells possessed a relatively large portion (34.84%) and the cell number density was the largest (6.07 × 10^5^/cm^3^).

## Figures and Tables

**Figure 1 polymers-14-03273-f001:**
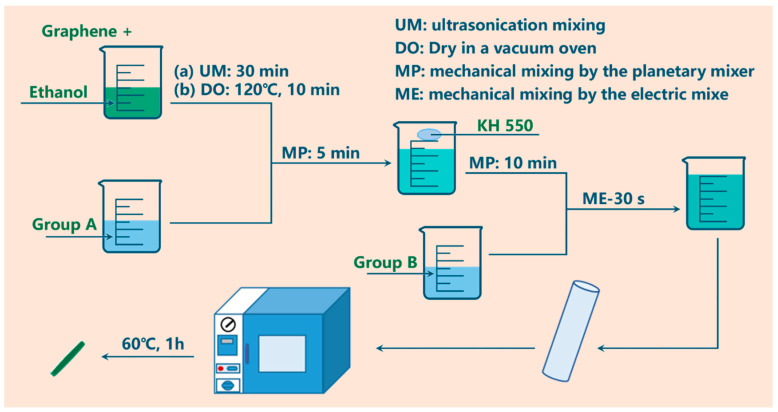
Preparation process of silicone rubber foam/graphene (SR/Gr) composites.

**Figure 2 polymers-14-03273-f002:**
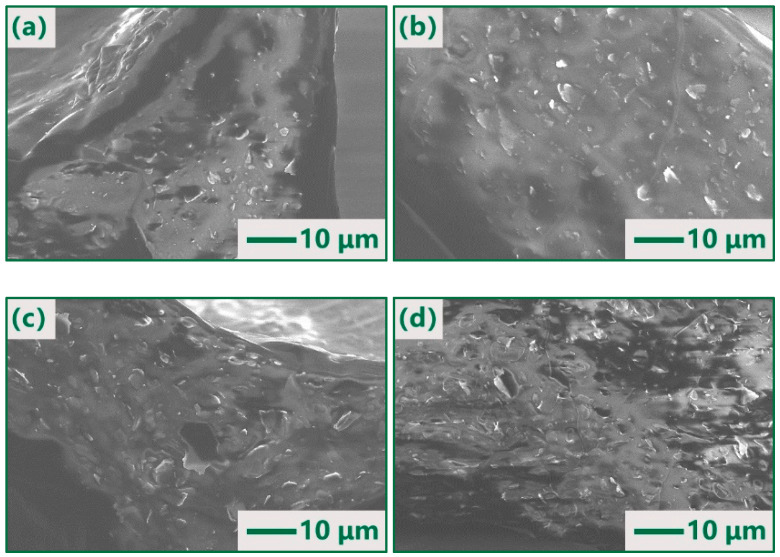
Morphology of SR/Gr composites (foaming expansion rate = 4) containing different Gr contents: (**a**) 1 wt%; (**b**) 2 wt%; (**c**) 3wt%; (**d**) 4 wt%.

**Figure 3 polymers-14-03273-f003:**
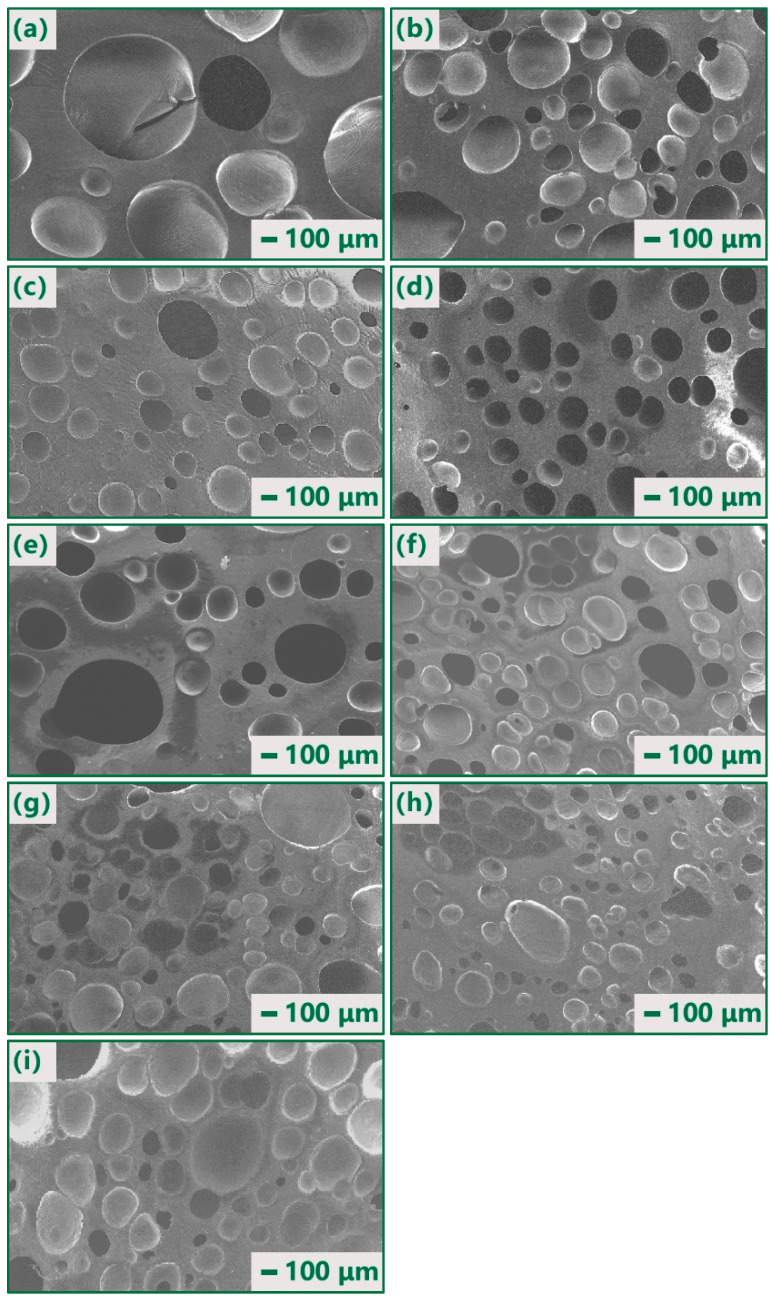
Cell structure of the composites of various expansion rates and Gr contents: (**a**) 0 wt%, F2; (**b**) 1 wt%, F2; (**c**) 2 wt%, F2; (**d**) 3 wt%, F2; (**e**) 4 wt%, F2; (**f**) 3 wt%, F3; (**g**) 3 wt%, F4; (**h**) 3 wt%, F5; (**i**) 3 wt%, F2-5.

**Figure 4 polymers-14-03273-f004:**
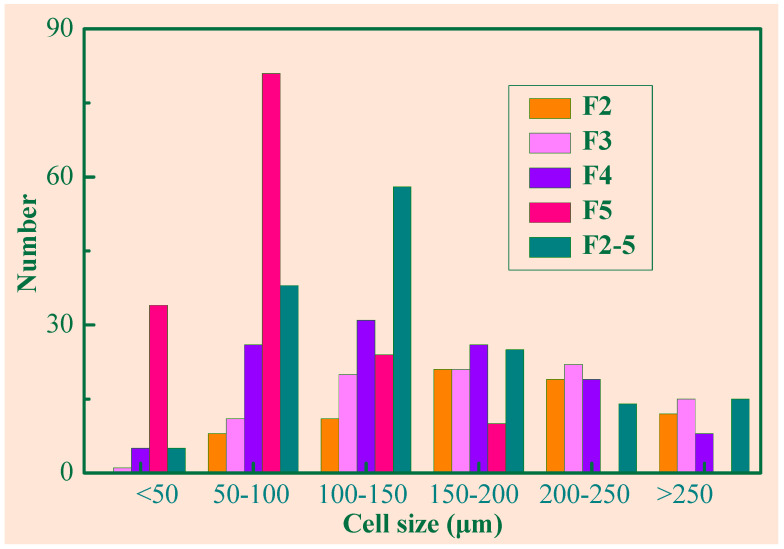
Cell size distribution of the SR/Gr composite films (1 wt% Gr).

**Figure 5 polymers-14-03273-f005:**
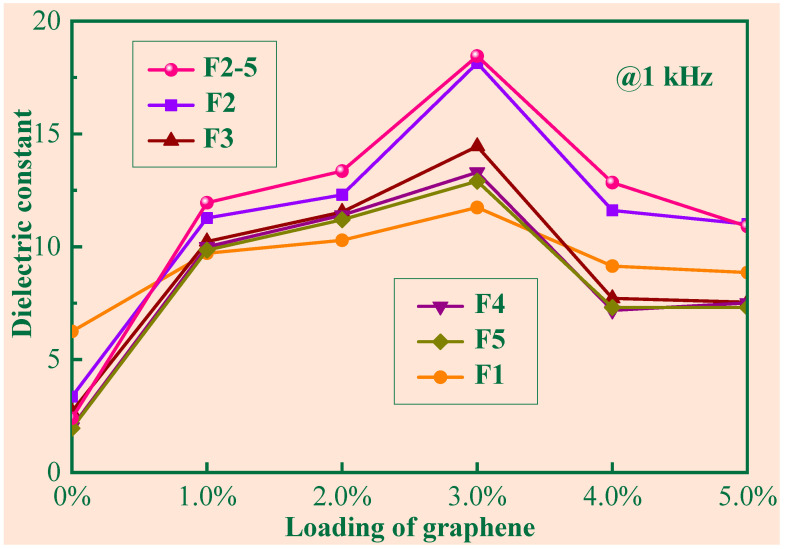
The dielectric constant (1000 Hz) of the composites.

**Figure 6 polymers-14-03273-f006:**
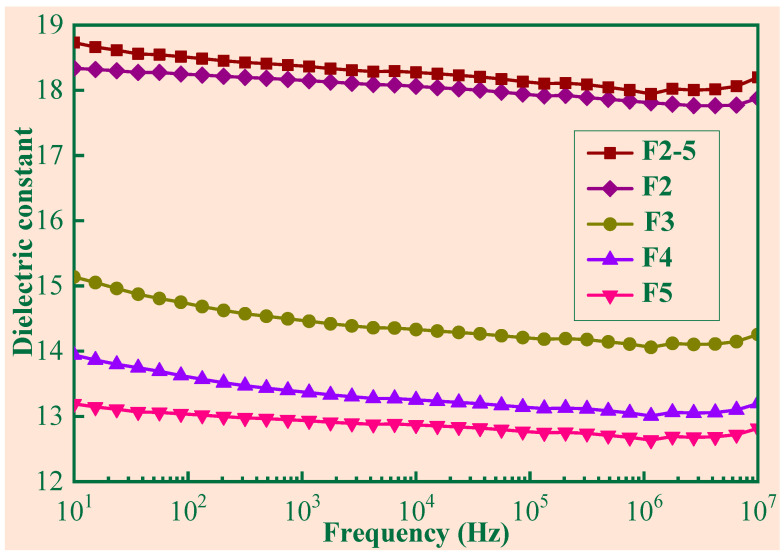
The dielectric constant of the SR/Gr(3 wt%) composites.

**Figure 7 polymers-14-03273-f007:**
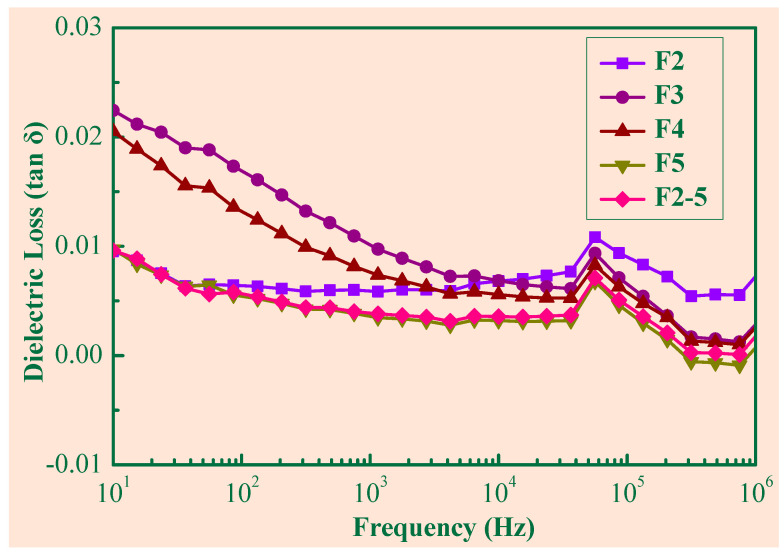
The dielectric loss of the SR/Gr composites (3 wt%).

**Table 1 polymers-14-03273-t001:** Experimental materials.

Foams	Mass Ratio	α
F1	HY-F661A:HY-F661B = 1:1	1
F2	HY-F662A:HY-F662B = 1:1	2
F3	HY-F663A:HY-F663B = 1:1	3
F4	HY-F664A:HY-F664B = 1:1	4
F5	HY-F665A:HY-F665B = 1:1	5
F2-5	HY-662A:HY-662B:HY-665A:HY-665B = 1:1:1:1	3.5

**Table 2 polymers-14-03273-t002:** Basic properties of SR/Gr (1 wt%) foam materials.

Sample	F1	F2	F3	F4	F5	F2-5
Void fraction *f* (%)	0	50	66.7	75	80	71.4
Average cell size (μm)		191	182	148	79.4	143
Cell number		71	90	115	149	155
<50 μm		0	1	5	34	5
50–100 μm		8	11	26	81	38
100–150 μm		11	20	31	24	58
150–200 μm		21	21	26	10	25
200–250 μm		19	22	19	0	14
>250 μm		12	15	8	0	15
Large cell (>150 μm)		73.24%	64.44%	46.09%	6.71%	34.84%
Cell density (10^5^/cm^3^)		2.13	3.36	5.49	7.15	6.07

**Table 3 polymers-14-03273-t003:** Dielectric constant of silicone rubber foams without graphene.

Foams	Void fraction *f* (%)	Theoretical ε*_r_*	Experimental ε*_r_*
F1	0	6.25	6.25 ± 0.25
F2	50	3.63	3.36 ± 0.31
F3	66.7	2.75	2.62 ± 0.10
F4	75	2.31	1.96 ± 0.32
F5	80	2.05	1.67 ± 0.28

## Data Availability

The data presented in this study are available on request from the corresponding authors.

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
