# Peer review of "Preparation of Novel Graphene/Silicone Rubber Nanocomposite Dielectric Foams"

_polymers, 2022, doi:10.3390/polym14163273_

Round 1

Reviewer 1 Report

The work «Preparation of novel graphene/silicone rubber nanocomposite dielectric foams» is devoted wide-spreading topic about nanocomposite with carbon filler. In the introduction, the authors well described the well-known fillers of various chemical nature for the studied matrix in the form silicone rubber, indicated the features and difficulties of obtaining composites based on it and the values of the achieved indicators, noted the novelty of the study. The text of the manuscript is successfully accompanied by high-quality figures and tables. The description of the methodology is quite detailed. The results obtained are well discussed. Conclusions are specific and generalizing. But I have some comments:

1. The y-axis for the graphs in figures 5 and 6 "dielectric constet" needs to correct this error.

2. For every equations, there should be an explanation for each component.

3. It is necessary to justify why only one and the simplest model was used to analyze the permittivity. It is not clear from the text what the dielectric constant simulation was used for. To justify the presence of a synergistic effect? There is no statistical processing of simulation results. There are no conclusions about the possible practical application of the study results.

4. The term "novel process" should be abandoned, this is not correct.

Reviewer 2 Report

In the manuscript "Preparation of novel graphene/silicone rubber nanocomposite dielectric foams" the authors reported on the preparation and dielectric behavior evaluation of some graphene/ silicone rubber nanocomposites. The paper is well written and the experimental results are good correlated. However, there are some minor issues to be corrected:

-in the last paragraph from Introduction, the author mentioned about dramatically increase of dielectric properties, lines 82-85, but the results revealed a decrease in dielectric constant (Table 2, Figure 6). Please explain and add some other literature comparative results.

- in Materials and Experiments, please add a table to explain the composition of the samples F1-F5, instead of the lines 109-112. 

-why did authors choose silicone rubbers with relative low molar mass?

-how did  they evaluate the foaming expansion rate? by SEM? Other methods should be used to confirm.

-I think that authors showed more of an influence of cell number and cell dimensions with dielectric behavior than with the graphene filler amount! Please explain why?

-did authors make a compatibility of graphene filler to avoid agglomeration? Please add these information!

Based on these comments, I consider that this paper would need some improvements before acceptance, so that I recommend Minor revision!

Round 2

Reviewer 1 Report

My comments have been corrected. I believe that the manuscript is ready for publication in Polymers.